# Effect of Strontium-Doped Bioactive Glass on Preventing Formation of Demineralized Lesion

**DOI:** 10.3390/ma14164645

**Published:** 2021-08-18

**Authors:** Lin-Lu Dai, May-Lei Mei, Chun-Hung Chu, Irene Shuping Zhao, Edward Chin-Man Lo

**Affiliations:** 1Faculty of Dentistry, The University of Hong Kong, Hong Kong; doreen07@hku.hk (L.-L.D.); chchu@hku.hk (C.-H.C.); 2Faculty of Dentistry, University of Otago, Dunedin 9016, New Zealand; may.mei@otago.ac.nz; 3School of Dentistry, Shenzhen University Health Science Center, Shenzhen 518000, China; zhao110@szu.edu.cn

**Keywords:** bioactive glass, fluoride, prevention, caries, strontium

## Abstract

This study investigated the effect of strontium-doped bioactive glass (SBAG) on the formation of dental demineralized lesions. Materials and methods: The study materials were 48 sound human tooth specimens with both dentine and enamel, divided equally into four groups: Group 1 (SBAG), Group 2 (SBAG+Fluoride), Group 3 (Fluoride), and Group 4 (Water as control). After 14 days of pH cycling, the surface morphology of the specimens was observed by scanning electron microscopy. Crystal characteristics of the precipitates were assessed by X-ray diffraction (XRD). Micro-CT was used to measure the mineral loss and the depths of the demineralized lesions formed. Results: Exposure of collagen in inter-tubular areas in dentine was seen in the control group (Group 4) but not in Groups 1 to 3. In Group 2, there were obvious granular particles on the surface of the dentine. XRD revealed precipitation of apatites on the surface of the tooth specimens in Groups 1 to 3. The mean lesion depths in dentine were 81.80 μm, 30.68 μm, 39.04 μm, and 146.36 μm in Groups 1 to 4, respectively (*p* < 0.001). Lesions in enamel were only found in the control group. The mean mineral loss values in the dentine lesions were 1.25 g/cm^3^, 0.88 g/cm^3^, 0.87 g/cm^3^, and 1.65 g/cm^3^, in Groups 1 to 4, respectively (*p* < 0.001). Conclusion: Strontium-doped bioactive glass has a preventive effect on the formation of demineralized lesions in enamel and dentine.

## 1. Introduction

Dental caries is a chronic and multifactorial disease induced by organic acids that dissolve the tooth minerals [1]. The acid diffuses through the dental plaque (a bacterial biofilm) on the tooth surface and dissolves hydroxyapatite which is the main mineral content of enamel and dentine, while saliva in the mouth can buffer acid, inhibit demineralization, and promote remineralization [2]. Dental caries is preventable, and the risk factors can be managed during the process through different approaches [1]. Fluoride, when present in oral fluid, can be adsorbed onto the tooth mineral crystal surface to form fluorapatite which is less soluble and prohibits the dissolution of tooth minerals during acid attack [3]. In addition, the use of topical fluorides, such as fluoride toothpaste, is effective for caries prevention [4].

The bioactive glass was first introduced for dental use in 2006 [5]. The most popular use, for example, the commercial bioactive glass Bioglass 45S5, is for treating dentine hypersensitivity [6]. When bioactive glass comes into contact with saliva in the mouth, it will induce a rise in pH and precipitations on the tooth surface, forming an apatite-like layer that blocks the dentine tubules and produces alleviation of hypersensitivity [7]. A review concluded that bioactive glasses can potentially enhance and promote enamel remineralization [8]. Another review reported that the effect of bioactive glasses on dentine remineralization is through stimulating apatite formation in dentine [9]. Bioactive glasses can also induce apatite deposition on the surface of carious lesions, forming an enrichment layer and leading to remineralization of the lesions [7]. Our recent investigation showed that bioactive glass can form a new apatite layer on the surface of demineralized enamel and dentine lesions to promote remineralization [10].

A recent development in bioactive glass is the incorporation of strontium into the material [11]. Strontium can replace calcium in the apatite lattice and has a weak antibacterial effect [12]. The incorporation of strontium in the crystal lattice can inhibit the dissolution of hydroxyapatite, the main inorganic component of enamel and dentine, which may be beneficial in preventing dental caries [7]. Our previous study on strontium-doped bioactive glass (SBAG) only focused on its remineralizing effect on demineralized lesions in enamel and dentine. So far, no study on the effect of SBAG on caries prevention has been reported. Thus, the aim of the present study was to investigate the effect of SBAG on the formation of demineralized lesions in enamel and dentine in an in vitro model.

## 2. Materials and Methods

### 2.1. Preparation of Specimens

Twelve extracted human molars were selected from the tooth collection of a teaching hospital (Prince Philip Dental Hospital) and stored in a 0.5% thymol solution at 4 °C before use. A slice containing both enamel and dentine was prepared from each tooth using a low-speed cutting machine (ISOMET 1000, Buehler, LakeBluff, IL, USA) under running deionized water. Each slice was cut into four specimens. Thus, 48 specimens in total were obtained. The specimens were polished using micro-fine 4000 grid papers and then washed by running deionized water. Subsequently, the specimens were examined under a stereomicroscope to exclude those with cracks or defects. In each specimen, only half of the upper surface was left uncovered while the other surfaces were covered by acid-resistant nail varnish. The study design is shown in Figure 1.

### 2.2. Assessment of the Changes in Enamel and Dentine

#### pH Cycling Method and Experimental Treatment

A pH cycling method was used to investigate the caries preventive effect of the SBAG and fluoride on sound enamel and dentine [13]. The exposed surface of each specimen was treated with the corresponding suspension or solution for 2 min according to its group allocation. Each of the four prepared specimens from the same tooth was randomly distributed into one of the four study groups, resulting in 12 specimens from 12 different teeth being allocated to each group. The SBAG powder used in this study was HX-BGC provided by Dencare (Chongqing, China) Oral Care Co. Ltd. In Group 1 (SBAG), the specimens were treated with a 5% (0.0526 g/mL) suspension of SBAG (0.263 g SBAG powder mixed with 5 mL deionized water). In Group 2 (SBAG+F), the specimens received treatment with a mixture of 5% SBAG suspension and 1450 ppm fluoride (0.0145 g NaF powder and 0.263 g SBAG powder added into 5 mL deionized water). In Group 3 (F), fluoride solution at a concentration of 1450 ppm (0.0145 g NaF powder dissolved in 5 mL deionized water) was applied on the surface of the specimens. Group 4 (Water) was the negative control group receiving an application of deionized water on the surface of the specimens. The suspensions and solutions were freshly prepared and vortexed for 20 s before each application. A micro-brush (Premium Plus International Ltd., Hong Kong, China) was used to apply the study agents on the exposed surface of the specimens, followed by waiting for 2 min. Then, the specimens were gently rinsed with a jet of deionized water to remove the applied agents. This preparation process took 30 min to complete for all 48 specimens. Afterwards, all the specimens were immersed in a demineralizing solution (1.5 mM CaCl_2_, 0.9 mM KH_2_PO_4_, 50 mM acetic acid, pH = 4.5) for 8 h. After the 8 h demineralization, the specimens were rinsed with deionized water and were applied with the four agents according to their group allocation. The specimens then went through a 15 h immersion in remineralization solution (1.5 mM CaCl_2_, 0.9 mM NaH_2_PO_4_, 150 mM KCl, pH = 7.0). All the specimens were subjected to pH cycling for 14 days.

### 2.3. Surface Morphology

Four specimens from each group were fixed in 2.5% glutaraldehyde overnight. They were ultrasonically cleaned three times with distilled water and then dehydrated using ethanol solutions with different concentrations (75%, 80%, 95%, and 100%). The dehydrated specimens were further dried in a critical-point dryer (Leica EM CPD300, Wetzlar, Germany) and sputter-coated with carbon. Afterwards, the morphology of the specimen surface was observed under a scanning electron microscope (SEM, Hitachi S-4800FEG scanning electron microscope, Hitachi Ltd., Tokyo, Japan).

### 2.4. Mineral Loss and Lesion Depth

X-ray micro-computed tomography (micro-CT) (SkyScan1172, SkyScan, Antwerp, Belgium) was used to measure the mineral loss and the depths of the newly formed demineralized lesions in enamel and dentine. The voltage and current settings were 80 kV and 100 μA, respectively. Five specimens from each group were scanned and NRecon reconstruction software (SkyScan, Antwerp, Belgium) was used to reconstruct the scanned specimen images. CTAn data analyzing software (SkyScan, Antwerp, Belgium) was used to view and analyze the reconstructed three-dimensional images of each specimen. Ten images of each specimen were randomly selected from the reconstructed cross-sectional images to measure the mineral loss (Δ mineral density) and the lesion depth.

### 2.5. Crystal Characteristics

Characteristics of the mineral crystals on the surface of the tooth specimen in each group (3 specimens per group) were assessed by an X-ray diffractometer (XRD, Rigaku SmartLab 9 kW, Tokyo, Japan) equipped with a CuKα lamp (λ = 1.54056 Å). The parameters for collecting data were: 2θ range = 20° to 60°, step size = 0.02° and scan speed = 0.6 s/step. To minimize error, the data collection process was repeated three times. The software Jade 6 (MDI, Livermore, CA, USA) was used to analyze the patterns to confirm the phases of the minerals in each study group.

### 2.6. Statistical Analysis

Data on lesion depths and mineral loss were assessed for normality by using the Shapiro–Wilk test. A one-way analysis of variance was used to compare the differences in lesion depths among the four study groups. IBM SPSS version 27 was used to conduct all the analyses. The statistical significance level adopted was 5%.

## 3. Results

Figure 2 shows the surface morphology of the transverse section of dentine in the four groups. In Group 1, a relatively smooth surface of dentine with little exposure of collagen fibers in the inter-tubular areas was observed (Figure 2a) and the structure of intra-tubular collagen fibers was intact (Figure 2b). In Group 2, there was no collagen fiber exposure in the inter-tubular area (Figure 2c) and in the intra-tubular area, large grains nearly fully covering the collagen fibers were observed (Figure 2d). In Group 3, there were irregular precipitates on the surface with sparsely exposed collagen fibers (Figure 2e), while the intact structure was conserved in dentine tubules (Figure 2f). In contrast, in the control group, there was a relatively rough inter-tubular surface (Figure 2g) and loosened collagen fibers (Figure 2h).

Figure 3 shows the cross-sectional surface of the dentine. In Groups 1 and 2, there was precipitation of particles either on the cross-sectional surface or intra-tubular areas. There were large particles adhered to the dentine surface and the dentine tubules were filled with some smaller granular grains (Figure 3a–d). In Group 3, there were small particles deposited on the surface with partial exposure of the dentine tubules (Figure 3e,f). In the control group (Group 4), no granular deposition on the surface was observed (Figure 3g) and there were some collagen fibers sparsely distributed in the dentine tubules (Figure 3h).

Table 1 lists the details of the depth and mineral loss of the lesions in enamel and dentine. The mean lesion depths of dentine in Groups 1 to 4 were 81.80 ± 12.39 μm, 30.68 ± 17.91 μm, 39.04 ± 17.99 μm and 146.36 ± 36.86 μm, respectively (*p* < 0.001). Group 2 and Group 3 had a mineral loss of 0.88 ± 0.15 g/cm^3^ and 0.87 ± 0.19 g/cm^3^, respectively, which were less than those in Groups 1 and 4 (*p* < 0.001). Besides, the dentine mineral loss in Group 1 (1.25 ± 0.19 g/cm^3^) was significantly less than that in Group 4 (1.65 ± 0.13 g/cm^3^) (*p* = 0.007). Enamel mineral loss only occurred in the control group with a mean mineral loss of 1.37 ± 0.62 g/cm^3^. Figure 4 and Figure 5 show the micro-CT images of the four groups. The demineralized lesion in Group 1 (Figure 4a) was shallower than that in Group 4 (Figure 4d). In Group 2 and Group 3, there was a precipitated layer on the lesion surface (Figure 4b,c).

Typical XRD patterns of the minerals in the specimens in the four study groups are shown in Figure 6. The pattern of scattered X-rays in Group 4 corresponds to the planes (002), (210), (211), and (300) which indicates apatite. Bragg’s reflections showed that the deposition on the surface of the specimens was apatite. The (211) and (300) reflections in the two groups with SBAG were sharper and higher than those in the other two groups, which may indicate that the crystalline degree of the precipitates in these two groups was higher than that in the fluoride group or the control group.

## 4. Discussion

Results of this study show that SBAG has a preventive effect on demineralized lesion formation in the sound enamel and dentine. It is known that when bioactive glass is exposed to water, the ions such as calcium and phosphate in the glass are released and lead to an elevation of the localized pH which can promote the precipitation of these ions to form an apatite-like layer on the tooth surface [7,14]. SBAG has a good antibacterial effect on cariogenic bacteria which also potentially helps to inhibit demineralization caused by the acid produced by bacteria [11]. The mechanical properties of bioactive glass can be improved by doping of strontium. A previous study reported that the incorporated strontium in the enamel can reduce the solubility of the minerals [15]. Strontium ions can partially or completely replace calcium ions to react with phosphate to deposit minerals on dental hard tissues [16]. Strontium-substituted hydroxyapatite and strontium-substituted fluorohydroxyapatite have higher bioactivity and stability than hydroxyapatite [17]. In addition, fluoride has been used in dentistry for over 70 years due to its effectiveness on caries control [18]. The mechanism of fluoride action on inhibition of demineralization is mainly through improving the resistance of minerals to dissolution when there is a pH fall [18]. The release of fluoride is higher in the presence of a strontium and fluoride-modified bioactive glass, which can decrease the dissolution of the newly formed apatite [19]. Furthermore, incorporation of both strontium and fluoride into hydroxyapatite lattice leads to a more stable structure with lower solubility [20,21].

The inter-tubular surface of the dentine in the treatment groups of the present study was partially covered with depositions, while the collagen network of the dentine in the control group collapsed after the pH cycling. This is consistent with the result of a previous study that, after treating with BAG-containing composite, crystals of various sizes were deposited on the dentine surface and partially occluded dentine tubules [22]. In another study, the reconstructed micro-CT images showed a reduced demineralization of dentine surface in the groups treated with fluoride products [23]. An in vitro study showed that the bioactive glass added into glass ionomer cement (GIC) can decrease the degree of enamel demineralization around orthodontic brackets [24], which is consistent with the results of the present study. An earlier study found that adding 45S5BAG fillers into sealants can prevent demineralization of enamel surfaces [25]. Besides, a type of fluorinated bioactive glass displayed an excellent inhibitory effect on the demineralization of enamel to prevent white spot lesions [26].

In the present study, XRD showed that apatite was formed in all the treatment groups. The analysis indicates that strontium and fluoride ions might have substituted calcium and hydroxyl ions in the apatite crystals. Strontium may be incorporated into the crystal lattice of the apatite precipitated on the dentine surfaces in Group 1 and Group 2. The deposited apatites in Group 2 and Group 3 might be also modified, with some hydroxyl ions replaced by fluoride ions in the apatite lattice. When the concentration of fluoride in solution exceeds 100 ppm, calcium fluoride (CaF_2_) will deposit and serve as a source for the formation of fluorapatite [17,18,27]. Therefore, in this study, possibly the precipitation on the surface of the dentine and enamel in Group 2 and Group 3 was fluorapatite instead of fluorohydroxyapatite. The intensity of the main peaks at around 31.66° and 32.74° (2θ) in Group 2 was higher than that of Group 3 and Group 1, and the two groups with SBAG had wider peaks than the other two groups. Thinner peaks in the XRD patterns of the groups with SBAG revealed increased crystallinity of the formed crystals as the substitution of strontium for calcium did not produce distortion in the apatite lattice [28]. The addition of some strontium can maintain the apatite structure and increase crystallinity [29,30].

The pH-cycling model used in this study was to simulate the process of alternations of demineralization and remineralization of dental tissues in the mouth. The composition of demineralization and remineralization solution includes the necessary ions such as calcium and phosphate ions that favor deposition of minerals and the rotating use of alkaline and acidic solution was to mimic the pH change in the dental biofilm [31]. Nevertheless, the limitation of this method is the lack of biological factors in this chemical model. Saliva contains certain proteins which can help remineralization [32], but the solutions in the present study did not contain proteins. Moreover, after a long period (14 days) of pH-cycling, the formation of demineralized lesions was the outcome of a continuous process that involved both demineralization and remineralization. Though the depths of the demineralized lesions in Group 1 were less than those of the control group, we cannot confirm the results were due to the effect of SBAG on inhibiting demineralization or on promoting remineralization of the demineralized lesions. Further studies should adopt a biological model to assess the preventive effect of SBAG on dental caries formation and clinical trials are recommended to examine the effectiveness of bioactive glass products on dental caries prevention.

## 5. Conclusions

The main results of the present study were that the lesion depths and mineral loss of the specimens treated with SBAG and fluoride were smaller than those of the control group. SEM images showed obvious precipitation on the dentine surface. XRD patterns illustrated the formation of apatite on the surface of the specimens. It can be concluded that the combined use of SBAG and fluoride has a synergistic effect on the prevention of demineralized lesion formation. In summary, strontium-doped bioactive glass has a potential preventive effect on the formation of demineralized lesions on sound enamel and dentine by reducing mineral loss and forming apatite on the tooth surface.

## Figures and Tables

**Figure 1 materials-14-04645-f001:**
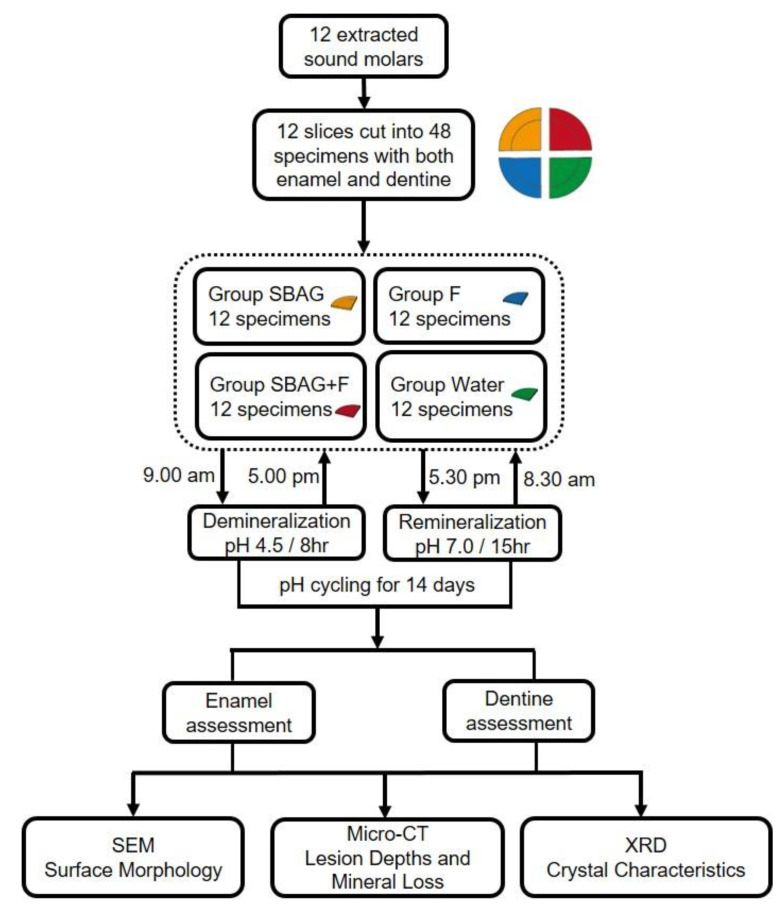
The flowchart of the experimental procedure.

**Figure 2 materials-14-04645-f002:**
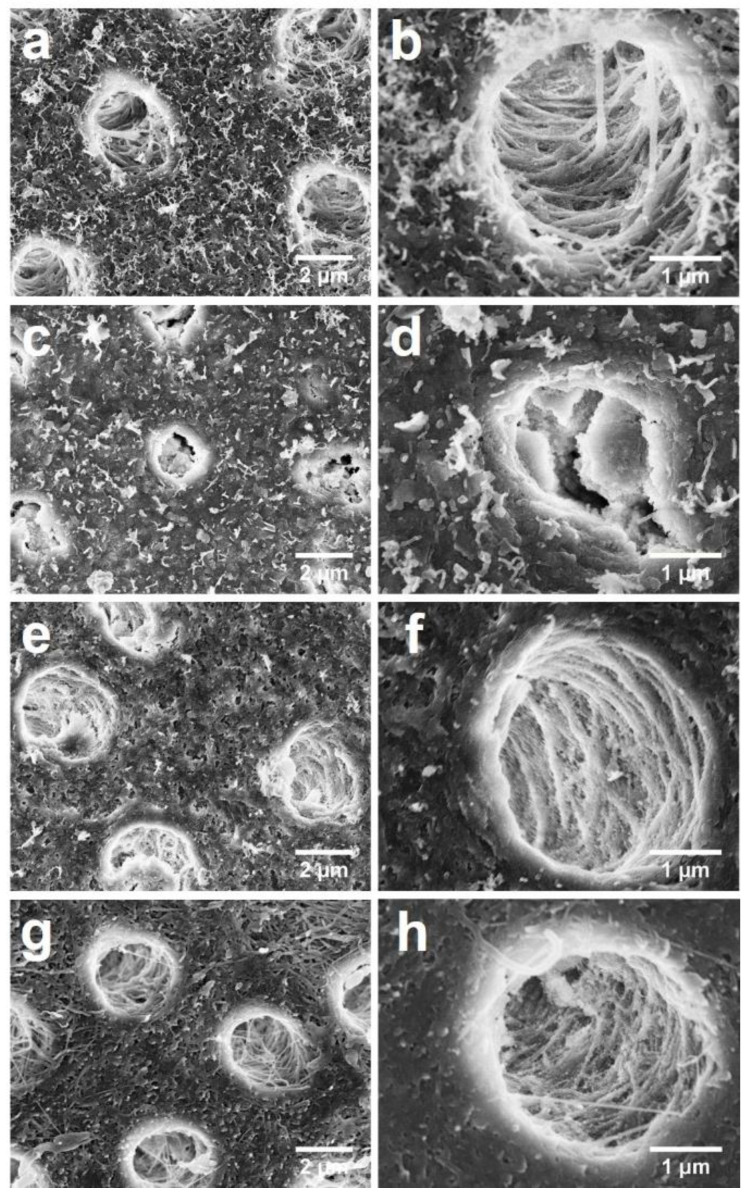
SEM images of the transverse section of dentine. (**a**) 8000× magnification view of Group 1 (SBAG); (**b**) 20,000× magnification view of Group 1 (SBAG); (**c**) 8000× magnification view of Group 2 (SBAG+F); (**d**) 20,000× magnification view of Group 2 (SBAG+F); (**e**) 8000× magnification view of Group 3 (F); (**f**) 20,000× magnification view of Group 3 (F); (**g**) 8000× magnification view of Group 4 (Control); and (**h**) 20,000× magnification view of Group 4 (Control).

**Figure 3 materials-14-04645-f003:**
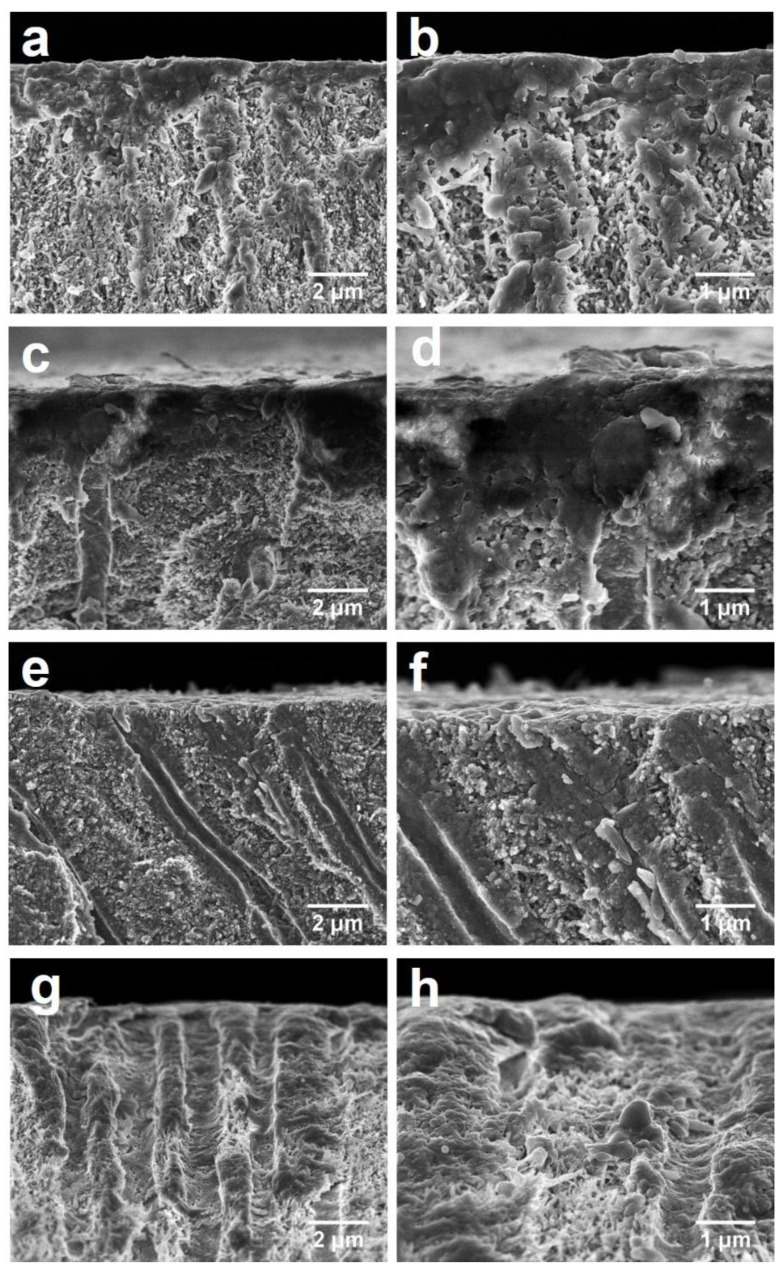
SEM images of the cross-sectional (longitudinal) surface of dentine. (**a**) 4000× magnification view of Group 1 (SBAG); (**b**) 8000× magnification view of Group 1 (SBAG); (**c**) 4000× magnification view of Group 2 (SBAG+F); (**d**) 8000× magnification view of Group 2 (SBAG+F); (**e**) 4000× magnification view of Group 3 (F); (**f**) 8000× magnification view of group 3 (F); (**g**) 4000× magnification view of Group 4 (Control); and (**h**) 8000× magnification view of Group 4 (Control).

**Figure 4 materials-14-04645-f004:**
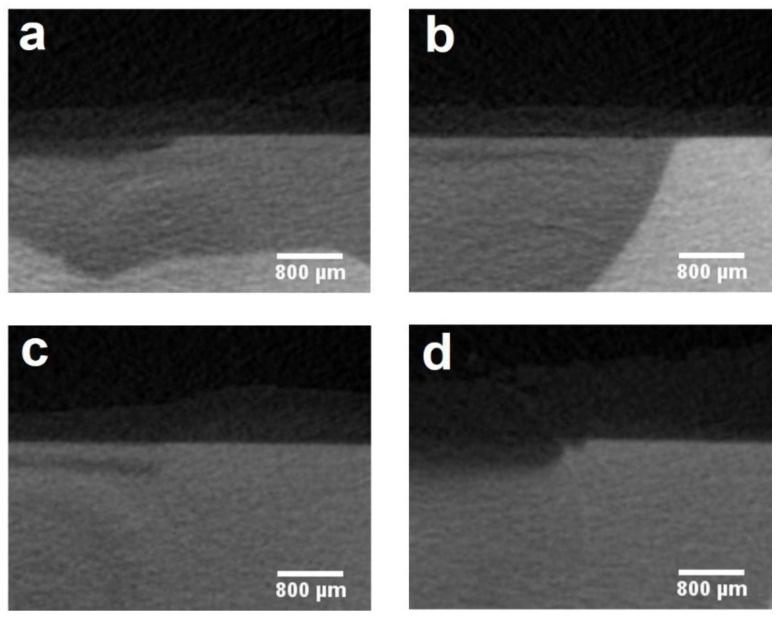
Micro-CT images of dentine in four groups: (**a**) Group 1 (SBAG); (**b**) Group 2 (SBAG+F); (**c**) Group 3 (F); (**d**) Group 4 (Control).

**Figure 5 materials-14-04645-f005:**
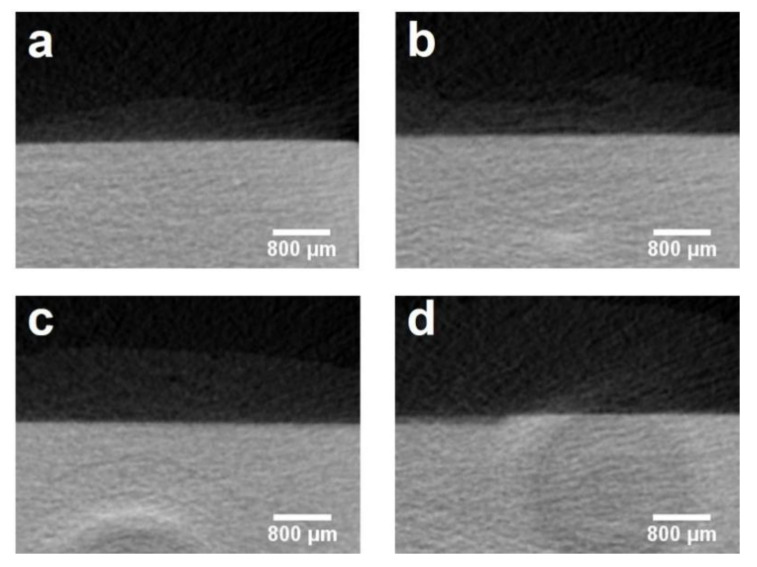
Micro-CT images of enamel in four groups: (**a**) Group 1 (SBAG); (**b**) Group 2 (SBAG+F); (**c**) Group 3 (F); (**d**) Group 4 (Control).

**Figure 6 materials-14-04645-f006:**
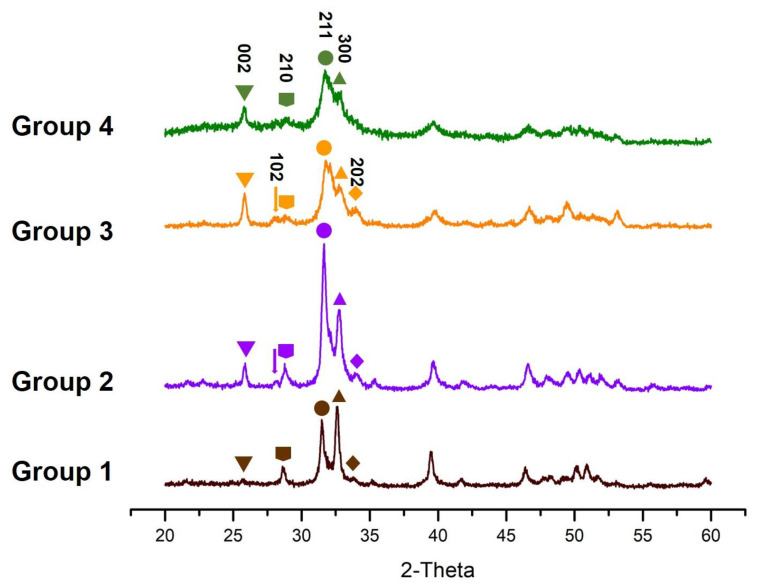
XRD patterns of the four groups: Group 1 (SBAG), Group 2 (SBAG+F), Group 3 (F) and Group 4 (Control).

**Table 1 materials-14-04645-t001:** The mean lesion depths and mineral loss of dentine and enamel after pH-cycling.

Group	Lesion Depth in Dentine (μm) (Mean ± SD)	Mineral Loss of Dentine (g/cm^3^) (Mean ± SD)	Lesion Depth in Enamel (μm) (Mean ± SD)	Mineral Loss of Enamel (g/cm^3^) (Mean ± SD)
SBAG (1)	81.80 ± 12.39	1.25 ± 0.19	0	0
SBAG+F (2)	30.68 ± 17.91	0.88 ± 0.15	0	0
F (3)	39.04 ± 17.99	0.87 ± 0.19	0	0
Water (4)	146.36 ± 36.86	1.65 ± 0.13	32.20 ± 24.76	1.37 ± 0.62
*p* vaule	*p* < 0.001	*p* < 0.001	-	-

## Data Availability

Publicly available datasets were analyzed in this study. This data can be found here: [https://datahub.hku.hk/].

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
