# Peer review of "Effect of Strontium-Doped Bioactive Glass on Preventing Formation of Demineralized Lesion"

_materials, 2021, doi:10.3390/ma14164645_

Round 1

Reviewer 1 Report

Dai et al. highlighted the effect of a strontium-doped bioactive glass on preventing the formation of demineralized lesions. The idea of the article is interesting and in general, the results sustain the aim of the article. Please, take into consideration the following remarks:

  1. Lines 39 – 50: The references 5-9 aren’t recent (2006-2019), so the authors must replace them with new ones, or should delete the words: recent, recently.
  2. The authors must mention the novelty of the study.
  3. The authors must mention the sources of the used reagents.
  4. Figure 6, containing XRD data, is missing. 
  5. Conclusions section is too short. Try to add at least 5-6 sentences with the main data obtained in the study.
  6. There are a lot of grammar and punctuation mistakes and also, English language needs proofreading.

Author Response

1. Lines 39 – 50: The references 5-9 aren’t recent (2006-2019), so the authors must replace them with new ones, or should delete the words: recent, recently.

Response:

Done. The words “recent” and “more recently” were deleted. The statements have been rephrased accordingly.

2. The authors must mention the novelty of the study.

Response:

Done. The novelty of this study is added to Introduction at Line 64-65 and 72-74.

3. The authors must mention the sources of the used reagents.

Response:

SBAG powder used in this study was provided by Dencare (Chongqing) Oral Care Co. Ltd, this is added to Line 103-105.

4. Figure 6, containing XRD data, is missing.

Response:

Done, Figure 6 is added.

5. Conclusions section is too short. Try to add at least 5-6 sentences with the main data obtained in the study.

Response:

Done. Conclusion has been enriched. (Line 312-320)

6. There are a lot of grammar and punctuation mistakes and also, English language needs proofreading

Response:

Done. The whole manuscript was carefully proofread and language mistakes have been corrected.

Reviewer 2 Report

"Effect of a new strontium-doped bioactive glass on preventing formation of demineralized lesion" by Lin Lu Dai, May Lei Mei, Chun Hung Chu, Irene Shuping Zhao and Edward Chin Man Lo reports analysis of strontium-doped bioactive glass (HX-BGC) and fluoride ions as important materials for the prevention of the formation of demineralized lesions. While the topic of the paper is of general interest and important, it does not present a dramatically new method or describes a new general principle that is different from what is already known.

My comments are:

1) The current abstract layout is as it was presented in J Dent, 308:103633 (2021). For example, in the current abstract the aspect (Materials and methods) highlights Forty-eight sound human tooth specimens distributed to four groups: Group 1 (HX-12 BGC), Group 2 (HX-BGC+F), Group 3 (F), and Group 4 (Water). In the J Dent, 308:103633 (2021) human tooth specimens were allocated as Group 1 received 5% HXBGC, Group 2 received 5% HX-BGC and 1450 ppm fluoride, Group 3 received 1450 ppm fluoride, and Group 4 received deionized water as negative control.

2) The introduction in many parts is confusing and is neither clearly described nor eloquently discussed to critically evaluate the manuscript in the framework of the pre-existing literature. In addition, as presented in this work, the aim and null hypothesis in the introduction are not different from the previous studies as covered in J Dent, 308:103633 (2021). For instance, the authors propose that “The aim of this study was to investigate the effect of HX-BGC on preventing formation of demineralized lesion on tooth surface in an in vitro model. The null hypotheses of this study was HX-BGC has no effect in preventing the formation of demineralized lesion in sound enamel and dentine”. Yet, in their previous published work in J Dent, 308:103633 (2021), the authors again proposed “Therefore, the present study aimed to investigate the remineralizing effect of a strontium-doped bioactive glass and fluoride ions on demineralized enamel and dentine. The null hypothesis was that the combined application of HX-BGC and fluoride would not enhance the remineralization of demineralized enamel and dentine”. Therefore, it is difficult to determine the novelty of the present experiments and the approach line of the present study and how it impacts scientific contribution.

3) The flowchart of the experimental procedure in Figure 1 is not different from previous work.

4) The results as presented in Figures 2-5 continue to demonstrate that new strontium-doped bioactive glass (HX-BGC) has a potential to prevent formation of demineralized lesions on enamel and dentine, as it was demonstrated previously in J Dent, 308:103633 (2021). The contribution is extremely marginal, in many parts confusing. For example, important data such as the diffraction data to differentiate between precipitated fluorapatite and fluorohydroxy-apatite on the surface of dentine and enamel in Group 2 and Group 3 was not presented in the manuscript.

5) The conclusion is not a true reflection of the entire manuscript.

Author Response

1) The current abstract layout is as it was presented in J Dent, 308:103633 (2021). For example, in the current abstract the aspect (Materials and methods) highlights Forty-eight sound human tooth specimens distributed to four groups: Group 1 (HX-12 BGC), Group 2 (HX-BGC+F), Group 3 (F), and Group 4 (Water). In the J Dent, 308:103633 (2021) human tooth specimens were allocated as Group 1 received 5% HXBGC, Group 2 received 5% HX-BGC and 1450 ppm fluoride, Group 3 received 1450 ppm fluoride, and Group 4 received deionized water as negative control.

Response:

We have rephrased the layout. Even the treatment groups are similar, the objectives of the current study is different from the previous one. We focus on the prevention effect on the sound human tooth rather than the arresting effect on existing lesion.

2) The introduction in many parts is confusing and is neither clearly described nor eloquently discussed to critically evaluate the manuscript in the framework of the pre-existing literature. In addition, as presented in this work, the aim and null hypothesis in the introduction are not different from the previous studies as covered in J Dent, 308:103633 (2021). For instance, the authors propose that “The aim of this study was to investigate the effect of HX-BGC on preventing formation of demineralized lesion on tooth surface in an in vitro model. The null hypotheses of this study was HX-BGC has no effect in preventing the formation of demineralized lesion in sound enamel and dentine”. Yet, in their previous published work in J Dent, 308:103633 (2021), the authors again proposed “Therefore, the present study aimed to investigate the remineralizing effect of a strontium-doped bioactive glass and fluoride ions on demineralized enamel and dentine. The null hypothesis was that the combined application of HX-BGC and fluoride would not enhance the remineralization of demineralized enamel and dentine”. Therefore, it is difficult to determine the novelty of the present experiments and the approach line of the present study and how it impacts scientific contribution.

Response:

The objectives of the current study is different from the previous one. We focus on the prevention effect on the sound human tooth rather than the arresting effect on existing lesion.

3) The flowchart of the experimental procedure in Figure 1 is not different from previous work.

Response:

Even the treatment groups are similar, the objectives of the current study is different from the previous one. We focus on the prevention effect on the sound human tooth rather than the arresting effect on existing lesion.

4) The results as presented in Figures 2-5 continue to demonstrate that new strontium-doped bioactive glass (HX-BGC) has a potential to prevent formation of demineralized lesions on enamel and dentine, as it was demonstrated previously in J Dent, 308:103633 (2021). The contribution is extremely marginal, in many parts confusing. For example, important data such as the diffraction data to differentiate between precipitated fluorapatite and fluorohydroxy-apatite on the surface of dentine and enamel in Group 2 and Group 3 was not presented in the manuscript.

Response:

The images in Figures 2-5 were taken on the specimens in the present study which were different from those in the previous paper on the remineralization effect of SBAG. A new figure (Figure 6) is added to the revised manuscript to show the XRD data obtained from the precipitates on the surface of dentine and enamel in the four study groups.

5) The conclusion is not a true reflection of the entire manuscript.

Response:

The conclusion has been rewritten (Line 312-320).

Reviewer 3 Report

The article is well prepared. The experiments and all tests are correctly planned and described.

Author Response

The article is well prepared. The experiments and all tests are correctly planned and described.

Response:

We thank the reviewer for the comments.

Round 2

Reviewer 1 Report

The manuscript has been improved, the authors addressing all my comments and recommendations. It can be published in present form.

Author Response

The manuscript has been improved, the authors addressing all my comments and recommendations. It can be published in present form.

Response: We thank the reviewer for the comments. 

Reviewer 2 Report

After carefully reading the revised version and the rebuttal of the manuscript "Effect of a new strontium-doped bioactive glass on preventing formation of demineralized lesion" by Lin Lu Dai, May Lei Mei, Chun Hung Chu, Irene Shuping Zhao and Edward Chin Man Lo, I have decided to recommend for its publication subject to addressing the points raised below.

My comments are:

(1) For the lines 49 – 56, the authors must include a sentence to differentiate the present manuscript to their previous publication as presented in J Dent, 308:103633 (2021).

(2) For lines 133 – 141 and 149 – 156, Figure 2'c and d' cannot be discussed before discussing 'a and b'. Similarly, Figure 2 'h' cannot be discussed before discussing 'g'. Either the figures should be reorganized, or the description should be altered.

(3) For lines 149 – 156, Figure 3 'g and h' cannot be discussed before discussing 'a, b, c and d'. Similarly, Figure 3 ' g and h' cannot be discussed before discussing 'e and f'. Either the figures should be reorganized, or the description should be altered.

(3) For lines 170 – 173, Figure 4 'b and c' cannot be discussed before discussing 'a'. Similarly, Figure 4 'd' cannot be discussed before discussing 'c'. Either the figures should be reorganized, or the description should be altered.

Author Response

(1) For the lines 49 – 56, the authors must include a sentence to differentiate the present manuscript to their previous publication as presented in J Dent, 308:103633 (2021).

Response: A sentence has been added to explain the difference between the present study (on prevention of new demineralized lesions) and the previous reported study (on remineralization of existing demineralized lesions) in lines 53-54.

(2) For lines 133 – 141 and 149 – 156, Figure 2'c and d' cannot be discussed before discussing 'a and b'. Similarly, Figure 2 'h' cannot be discussed before discussing 'g'. Either the figures should be reorganized, or the description should be altered.

(3) For lines 149 – 156, Figure 3 'g and h' cannot be discussed before discussing 'a, b, c and d'. Similarly, Figure 3 ' g and h' cannot be discussed before discussing 'e and f'. Either the figures should be reorganized, or the description should be altered.

(3) For lines 170 – 173, Figure 4 'b and c' cannot be discussed before discussing 'a'. Similarly, Figure 4 'd' cannot be discussed before discussing 'c'. Either the figures should be reorganized, or the description should be altered.

Response: The order of the related sentences in the text has been changed.